# Eulerian and Lagrangian Comparison of Wind Jets in the Tokar Gap Region

**Larry J. Pratt** [1,*]**, E. Jason Albright** [2]**, Irina Rypina** [1] **and Houshuo Jiang** [1]

[1]  Woods Hole Oceanographic Institution, Woods Hole, MA 02543, USA; irypina@whoi.edu (I.R.); hsjiang@whoi.edu (H.J.)
[2]  Los Alamos National Laboratory, Los Alamos, NM 87545, USA; ejalbright@lanl.gov
[*]  Correspondence: lpratt@whoi.edu

**Abstract:** The Lagrangian and Eulerian structure and dynamics of a strong wind event in the Tokar Gap region are described using a Weather Research and Forecasting (WRF) model hindcast for 2008. Winds in the Tokar Gap reach 25 m s$^{-1}$ and remain coherent as a jet far out over the Red Sea, whereas equally strong wind jets occurring in neighboring gaps are attenuated abruptly by jump-like hydraulic transitions that occur just offshore of the Sudan coast. The transition is made possible by the supercritical nature of the jets, which are fed by air that spills down from passes at relatively high elevation. By contrast, the spilling flow in the ravine-like Tokar Gap does not become substantially supercritical and therefore does not undergo a jump, and also carries more total horizontal momentum. The Tokar Wind Jet carries some air parcels across the Red Sea and into Saudi Arabia, whereas air parcel trajectories in the neighboring jets ascend as they cross through the jumps, then veer sharply to the southeast and do not cross the Red Sea. The mountain parameter *Nh/U* is estimated to lie in the range of 1.0–4.0 for the general region, a result roughly consistent with a gap jet having a long extension, and supercritical flows spilling down from higher elevation passes. The strong event is marked by the formation of a feature with a vertical cellular structure in the upstream entrance region of the Tokar Gap, a feature absent from the more moderate events that occur throughout the summer. The cell contains descending air parcels that are fed into the Tokar Gap and one of the neighboring gaps. An analysis of the Bernoulli function along air parcel trajectories reveals an approximate balance between the loss of potential energy and gain of internal energy and pressure, with surprisingly little contribution from kinetic energy, along the path of the descending flow. The winds in all gaps attain the critical wind speed nominally required to loft dust into the atmosphere, though only the Tokar Gap has a broad, silty delta region capable of supplying particulate matter for dust storms.

**Keywords:** coastal wind jets; Red Sea; Lagrangian pathways; hydraulics; dust storms; hydraulic jumps

---

## 1. Introduction

The Red Sea is fringed along much of its coastline by low mountains, including the Asir Range in Saudi Arabia and the Red Sea Hills in northeastern Sudan, all of which are thought to channel the prevailing winds along the axis of the Red Sea [1]. However, both ranges are punctuated by gaps and passes, and these have been linked to strong, localized offshore wind jets and funneled onshore winds [2]. The best known example of an offshore jet occurs in the summer, when the southwest monsoon blows through Sudan's Tokar Gap (Figure 1). The Tokar Wind Jet (hereafter TWJ) achieves wind speeds upwards of 25 m s$^{-1}$ and is associated with summer dust storms (Figure 2) and enhanced localized surface stresses and eddy generation in the Red Sea [3]. Based on a 30 year wind/wave climatology, [4] concluded that the highest surface waves in summer are generated at the center of the

Red Sea and are a consequence of the TWJ. Observations and model results [5,6] suggest that the Tokar Gap is part of an inland conduit for Indian Ocean monsoons, delivering a significant summer source of atmospheric moisture to the southern Red Sea and thence to the East African Highlands.

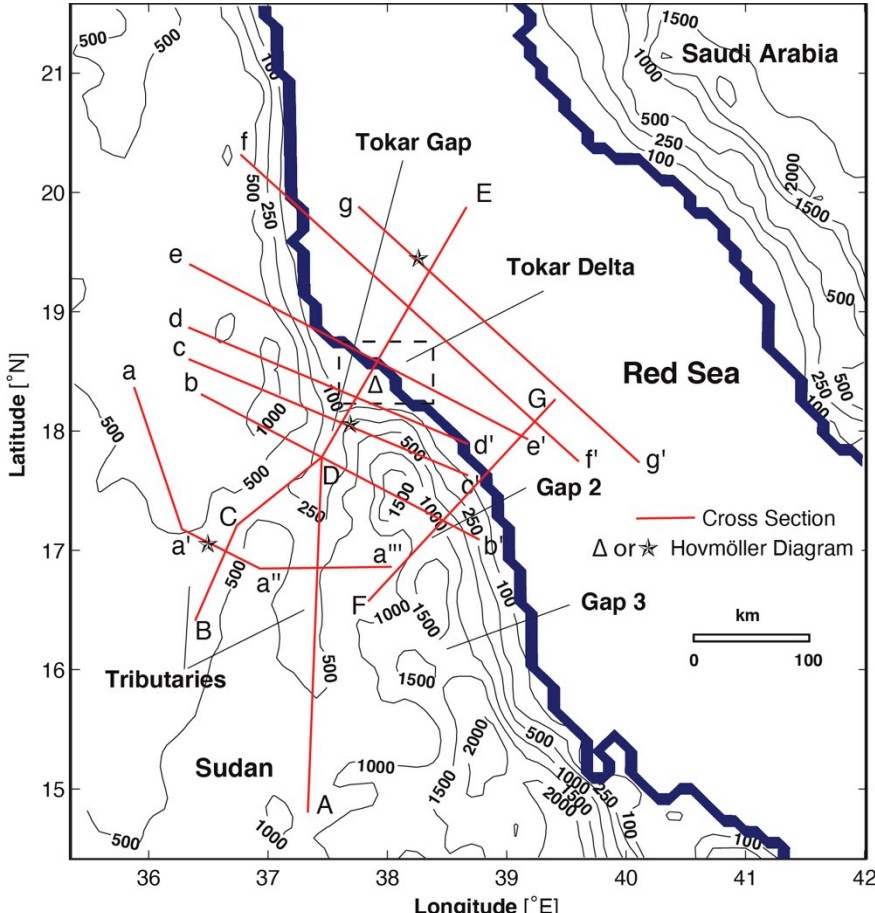

**Figure 1.** Regional map showing Red Sea Hills, Tokar Gap and Gaps 2 and 3 to the south. 'Tributaries' refer to the two entrance channels. The red lines show section locations, and the symbols indicate locations for Hovmöller diagrams, as cited in later figures. Topographic elevation contours are labeled in meters.

Model simulations indicate the presence of other wind jets that form in gaps in the Red Sea Hills and Asir Mountain Range [2,6]. We are primarily concerned with summer winds that form in the Tokar Gap and neighboring gaps, especially two unnamed gaps to the south (labeled Gap 2 and Gap 3 in Figure 1). We will sometimes refer to the latter as the secondary gaps and the wind events that occur within them as the secondary jets. Although the wind speeds in the secondary gaps reach or exceed the 25 m/s seen in the TWJ, the associated wind jets do not have nearly the same downstream reach as the TWJ. This difference is relevant to the local climate in that the monsoon moisture carried northeastward by the secondary jets does not penetrate all the way across the Red Sea and into Saudi Arabia.

Gap winds occur when air is driven from high to low pressure through the passes or cols in a mountain ridge. The funneling effect can produce low-level jets that extend well downstream of the ridge, an effect that can be enhanced in coastal areas due to the reduced drag coefficient over water. Coastal gap winds contribute significantly to local circulation patterns, extreme weather events, atmospheric transport, and the generation of ocean eddies. Sites of prominent gap winds in coastal regions include the Strait of Juan de Fuca [7–9], the Gulfs of Tehauntepec and Papagayo [10–12], the Columbia River Gorge [13], the Dinaric Alps [14] and the straits and gaps in the Philippine

Archipelago [15,16]. There are also many non-coastal examples [17]. Eddies spun up by the Tehuantepec gap winds can influence the annual cycle of sea surface temperature (SST) in the eastern Pacific warm pool [18]. Models suggest that ocean dipoles spun up by gap winds in the Philippine Archipelago can strip nutrient-laden waters from the coast and transport the nutrients far offshore [16].

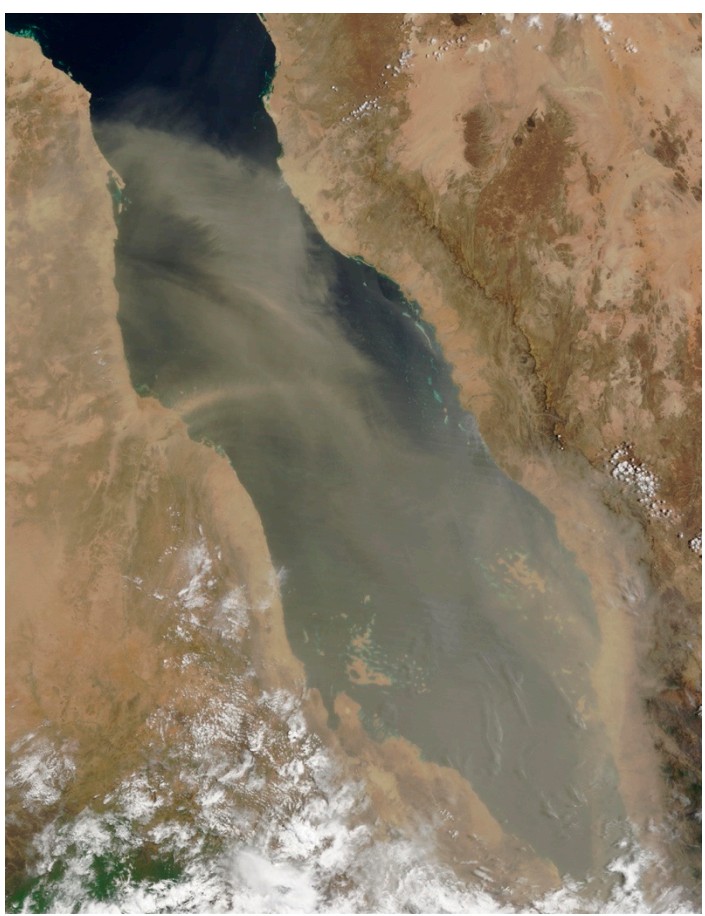

**Figure 2.** MODIS image taken at 8:00 UTC on 11 July 2002 showing dust plume associated with northwestward flow through the Tokar Gap and other gaps in the neighboring Red Sea Hills. (Courtesy of NASA: see http://visibleearth.nasa.gov).

The driving of air from high pressure to low pressure across the Red Sea Hills is evident in a regional simulation (Figure 3) using the Weather Research and Forecasting (WRF) model, the model used in this study. Within the topographic gaps, the 10 m winds (red arrows) cut across the (blue) contours of constant sea level pressure. The fields shown are the means for July 2008 and the wind jets that form are not as distinct as they would be in instantaneous examples (e.g., Section 3.3 below) but the mean TWJ can clearly be seen near 18° N and the mean expressions of several other gap flows can be seen to the south. Figure 1 also shows some topographic characteristics that set the Tokar Gap apart from other gaps in the region. To begin with, the topography is more typical of ravine than a mountain pass or col, with elevations dropping approximately 500 m from the interior Sudanese plateau to the Tokar Delta over a distance of about 400 km. Peaks to the immediate north and south of the Tokar Gap lie at 1000–1500 m elevations. By contrast, Gaps 2 and 3 contain topographic passes or cols at 1360 and 1430 m elevation and these potentially block lower level flow. At the head of the Tokar Gap lie two tributary canyons (hereafter the east and west entrance channels) that merge to form the main canyon (Figure 1). The gap then descends through its narrowest width of about 100 km before reaching the Tokar Delta, a rich alluvial plain formed by the flooding of the Baraka River and extending 50 km to the Red Sea and 80 km in either direction along the coastline. Both the Arabian and African coasts are

sources of silt for dust storms [19]. The Sudanese coast immediately around the Tokar Gap delta has been identified [20] as one of two major source regions for silt for Northern Hemispheric dust storms (Figure 2). By contrast, Gaps 2 and 3 terminate closer to the coast and do not have broad deltas.

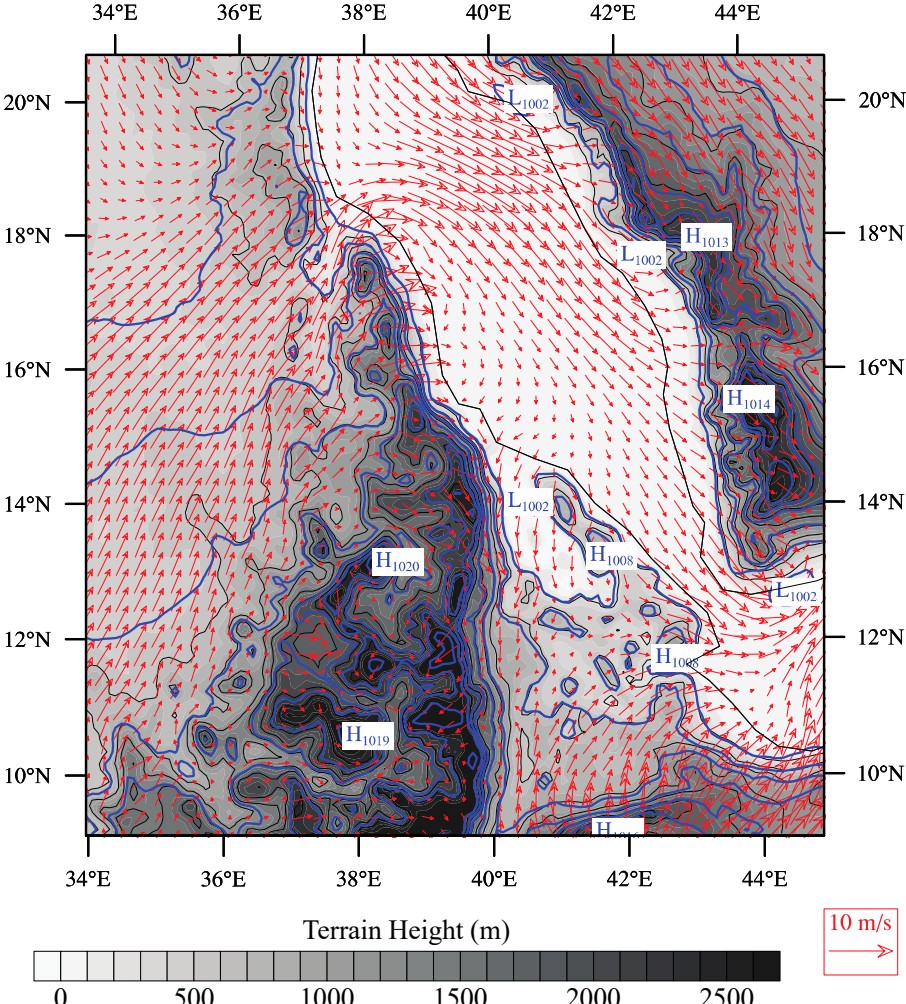

**Figure 3.** Mean 10 m wind vectors and sea level pressure (blue contours) for July 2008, from the Weather Research and Forecasting (WRF) model. The terrain height is shown in grey.

A feature that distinguishes the TWJ and neighboring gap jets from many other coastal gap winds is their strong diurnal variability. Jets were present on a nearly daily basis from mid-June to mid-September in a 2008 hindcast [2]. There is a strong daily cycle, with winds typically peaking around 04–06 UTC (7–9 a.m. local time) and maintaining a high directional consistency. The Red Sea also experiences a strong land/sea breeze [21], but the phasing can be slightly different from that of the TWJ. A possible influence [6] is the diurnal variation of the intertropical discontinuity, the leading edge of the southern monsoon air mass that feeds the TWJ. The elevated moisture content of the jets constitutes another distinguishing feature. During the week-long simulation analyzed by [6], the elevated humidity levels in the TWJ and neighboring jets led to significant pulses of moisture into the southern Red Sea region.

Our primary purpose was to compare and contrast the downstream reach of the TWJ and secondary jets, and to identify key dynamical processes that account for the differences. Included in the analysis will be maps of air parcel pathways at different levels for the three jets as a way to provide a comparison in terms of upstream origins and downstream destinations. This Lagrangian analysis also allows for the quantification of the energy transformations that occur along pathways. To set

the stage for these analyses, we present information regarding the overall Eulerian structure, time dependence, and hydraulic transitions characteristic of a wind event that occurred on 11–12 July 2008 and that is the central focus of our paper. The analysis was done using the same regional model output that was analyzed by [6]. A key element that will emerge is that the vertical thickness of the jets in the neighboring gaps is less than that of TWJ, whereas the peak winds are at least as large, causing them to be hydraulically supercritical and therefore subject to hydraulic jump formation. These features are described in Section 3 along with other relevant properties. Differences in the Eulerian properties also lead to differences in Lagrangian characteristics, in terms of the rate of stirring of air parcels, in the energy conversions that take place along parcel trajectories, and in the geographic origins and destinations of parcels, all discussed in Section 4. A distinctive feature with a vertical cellular structure that is triggered in the upstream reaches of the Tokar Gap at the outset of the wind event and that propagates westward as the event unfolds is also described in Section 4. Section 5 briefly explores some related issues, including comparison with other strong wind events in the region and conditions for lofting dust into the atmosphere.

## 2. Model Overview

Our results are based on a 14 month run of the Weather Research and Forecasting (WRF) model, version 3.0.1.1, with a 10 km horizontal resolution Red Sea subdomain nested within a 30 km resolution domain covering most of the Middle East, for which the 1° NCEP Global Final Analysis was used as initial and boundary conditions. The model employs 35 vertical levels, uses terrain-following eta coordinates, and produces output at 1 h intervals. Daily re-initializations occur each 36 h with 12 h overlap between consecutive runs. The first 12 h of each run is a spin-up period for which the data are discarded. The analysis is done for the day using the post spin-up 24 h data. Wind events normally last for 12 h or less and are usually captured continuously in time. Further details can be found in [22,23]. The model run was originally produced by [2], who describe validation based on surface wind speed from an air/sea buoy moored off the Saudi coast [24]. Further validation was described by [6] who compared the model output with radiosonde soundings of temperature, wind speed and water vapor at six observation stations located in Saudi Arabia and Egypt. Comparisons with vertically smoothed profiles were generally found to be good. A caveat is that the authors were unable to find in situ data within the Tokar Gap, or along the neighboring Sudan coast. For more details on the model validation, the reader is referred to [6].

## 3. Eulerian Structure

### 3.1. Diurnal Variability

The strong diurnal character of the gap winds, which typically begin around 00 UTC (3 a.m. local time) and terminate around 12 UTC has been documented [2,6]. The strongest wind event in the 2008 WRF model hindcast began late at night on 11 July and continued through the morning and into the early afternoon of 12 July. We now consider the temporal, structural and dynamical aspects of the TWJ along with the winds in Gaps 2 and 3 during this time period. The three-dimensional structure and evolution are revealed through data plotted at the stations and along the section lines indicated in Figure 1.

A view of the temporal evolution of the wind field over the 24 h period of 12 July 2008 is provided by Hovmöller plots showing the horizontal velocity as a function of elevation and time (Figure 4). The three plots were made at successive locations proceeding downstream: near the upstream entrances of the Tokar Gap (Figure 4a), in the narrowest part of the Tokar Gap (Figure 4b), and out over the Red Sea (Figure 4c). All locations are indicated by stars in Figure 1 and each was selected to lie in the high velocity core of the flow. The velocity contours indicate the magnitude of horizontal velocity directed along the axis (or 'thalweg') BCDE that begins in the west entrance channel of the Gap.

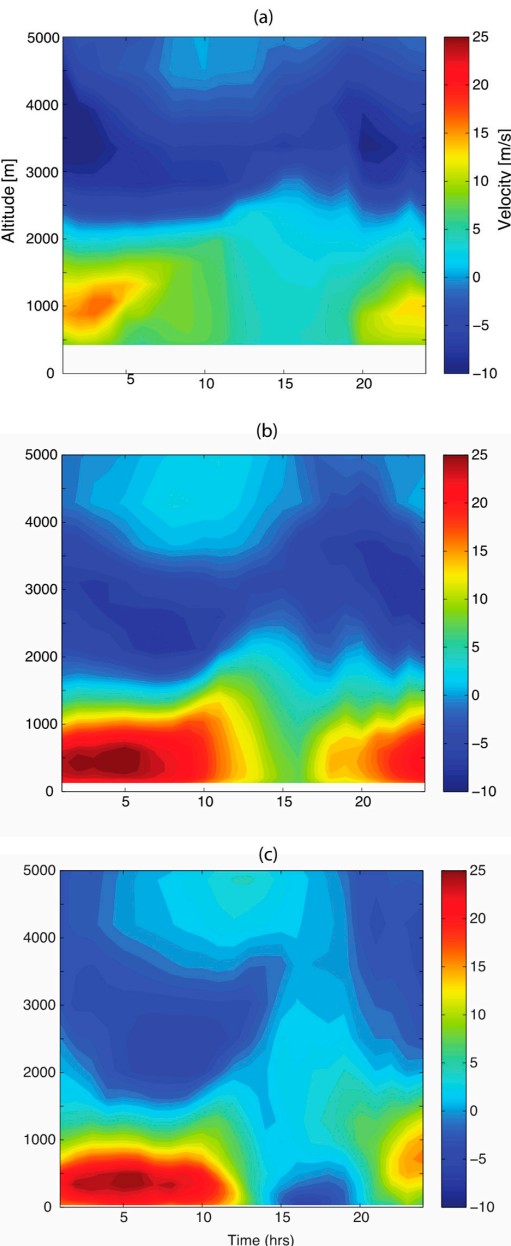

**Figure 4.** Hovmöller diagrams for the 12 July 2008 event. Shown is the along-thalweg component of horizontal velocity vs. time and elevation above the mean sea level at (**a**) the head of the northern spillway; (**b**) in the narrowest portion of the gap; and (**c**) out over the Red Sea. The locations of the stations are indicated by stars in Figure 1, with (**a**) lying close to point *a′*, (**b**) lying just northeast of *D*, and (**c**) lying near *E*. Time is UTC (local Saudi time minus 3 h). The upper boundaries of the white space at the bottoms of the panels indicates the elevation of the land.

The evolution below 2000 m shows a strong diurnal character, with high velocities during the night and early morning and nearly stagnant conditions during the late afternoon. The down-canyon winds extend up to 1500–2000 m above ground level and are generally weakest near the channel entrances, strongest at the narrowest section (Figure 4b), and still strong at the Red Sea section (Figure 4c). The downslope winds relax at around 12 UTC but remain positive in the gap itself. At the Red Sea location, the surface winds are eventually overcome by the opposing sea breeze and reverse direction at about 14 UTC. It is notable that the velocities in Figure 4c, which are predicted at a location nearly midway across the Red Sea, are nearly as large as those in the narrows (Figure 4b). A striking feature of all plots is the reverse flow that develops aloft, roughly between 1500 and 3500 m, and is strongest when

the jet is active. Figure 5c,f of [14] document a similar wind reversal during the strong phase of a bora. Similar reversals are present in models that reproduce the entrainment of overlying fluid into the wind jet, a process enabled by the formation of shear instabilities near the top of the jet. The entrainment sets up a reverse flow in which the fluid is drawn from the region aloft and further downslope. A model example is presented by [25], though the reverse flow is very weak. Flow reversals aloft of downslope winds were observed in other models (e.g., [26], Figure 7; [27], Figures 4 and 5) though they are associated with breaking structures that are not apparent in Figure 4.

### 3.2. Longitudinal Structure

Insights into the along-axis thermal and velocity structure of the Tokar Gap flow can be gained from the Figure 5 plots of potential temperature and along-axis velocity along the piecewise linear paths ADE and BCDE, as indicated in Figure 1. The plots show data at 05 UTC, corresponding to peak winds in the downstream reaches of the gap. The overall longitudinal picture is one of spilling and acceleration through the compound gap, with the highest velocities of approximately 20 m/s occurring downstream of the junction (D), where the gap narrows and then opens up over the delta. There is also some evidence of a decrease in static stability above the outflow: for example, the vertical displacement between the 315 K and 318 K potential temperature surfaces increases from 1000 m to 2000 m in the downstream direction. Another notable feature in the thermal structure is a slight increase in ground level potential temperature where the winds descend over land, followed by a decrease where the winds blow out over the cooler Red Sea.

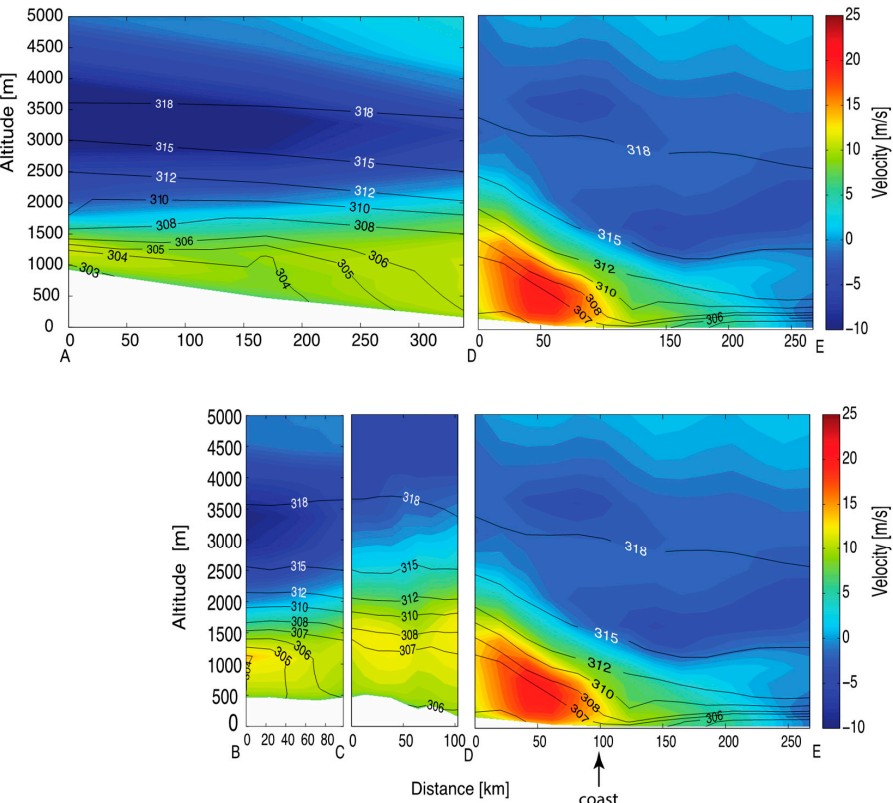

**Figure 5.** Longitudinal sections of potential temperature (black contours) and the along-section velocity component (colors) beginning in eastern (top) and western (bottom) spillways, ADE and BCDE respectively. The orientation of the section line is constant within each panel but changes between panels: thus, the along-section velocity component is discontinuous. The data are plotted at 05 UTC on 12 July 2008 and the section locations are shown in Figure 1.

Comparison with the winds in Gap 2, plotted at 05 UTC in Figure 6, reveals larger peak winds (≅25 m/s) along the downslope, and in the region extending from the pass near 80 km to the Red Sea shoreline near 165 km. However, the strong winds terminate about 10 km offshore. Whether one takes the 312 or 315 K potential temperature contour as an upper boundary for the descending flow or not, the vertical thickness (700–1000 m at the 125 km mark) is somewhat smaller than the thickness of the downslope wind layer in the Tokar Gap. The 312 K surface rises abruptly downstream of the core of strongest downslope winds, whereas the same surface continues to descend (Figure 5) where the Tokar Gap winds run out over the Red Sea.

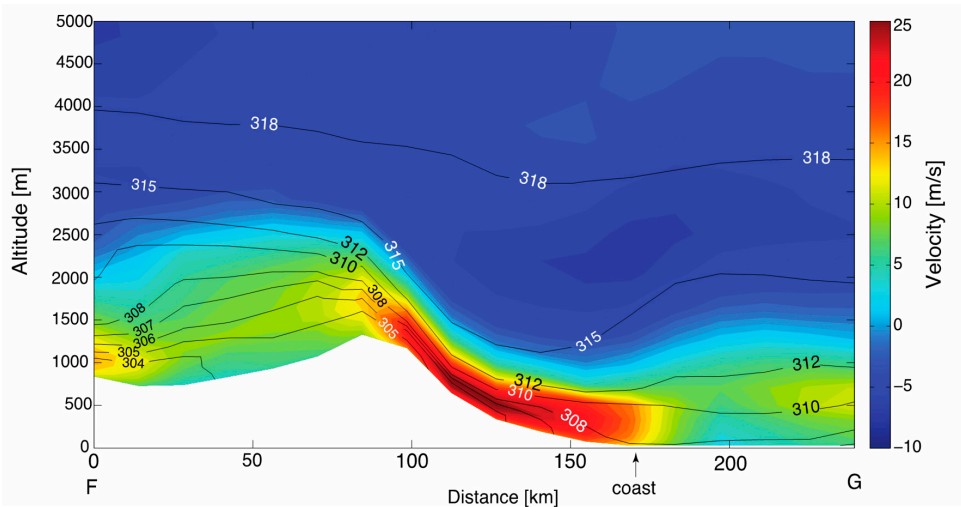

**Figure 6.** Horizontal, along-section wind velocity (colors) and potential temperature (contours) along the path FG of the downslope flow through Gap 2 (indicated on Figure 1) at 05 UTC on 12 July 2008.

## 3.3. Three Dimensional Structure

For gap wind applications, the branching, ravine-like topography of the Tokar Gap is less typical than the topographies of Gaps 2 and 3. For this reason, we examined the three-dimensional structure of the Tokar Gap winds further using a series of cross-sections (Figure 7a–g), plotted at 05 UTC and proceeding from upstream to downstream. Each section shows potential temperature and the normal component of horizontal velocity, with the viewer facing downstream. The section composited from *aa'*, *a'a''*, and *a''a'''* (Figure 7a) lies upstream and crosses both approach channels. Separate cores of moderately high velocity, with values up to 15 m s⁻¹, can be seen on either side of the small ridge (near 10 km in the right-hand panel) that separates the channels. At section *bb'* (Figure 7b), which is located at the junction of the approach channels, the separate jets merge to form a single core. The section extends from the northwest and southeast (left-to-right) across the bounding ridges. A high-speed core of flow can be seen along the sloping terrain to the south, across the ridge located near 200 km, revealing the presence of strong winds in this neighboring gap. Section *cc'* (Figure 7c), which cuts across the narrowest part of the gap, shows even higher velocities concentrated in a single core that fills the Tokar gap but continues into Gap 2 to the southeast. Section *dd'* cuts across the delta, *ee'* lies near the shoreline, and *ff'* and *gg'* lie mostly offshore (Figure 7d–g). The narrow (150 km wide), high velocity (25 m s⁻¹) core continues through *ff'*, diminishing somewhat by *gg'*. At section *ee'*, there are shallower, secondary regions of high velocity to the left and right (northwest and southeast) of the TWJ core. In the region lying between 0 and 100 km in Figure 7e the high winds are due to air spilling down the slope of the ridge that lies immediately to the north of the Tokar Gap. The weaker core to the south (between 300 and 350 km) consists of flow from Gap 2, which lies to the south of the Tokar Gap.

The surface expression of all three jets (Tokar Gap and Gaps 2 and 3) during maximum intensity (05 UTC) are apparent in maps of 10 m winds (Figure 8a) and surface potential temperature (Figure 8b). Winds in Gaps 2 and 3, and on neighboring downslopes, experience an increase in surface potential

temperature as they spill down over land, then undergo a decrease in potential temperature as they leave the coast and move offshore. By contrast, the TWJ maintains a low potential temperature as it crosses the delta and moves out over water. The disproportionally large reach of the TWJ compared to the jets in Gaps 2 and 3 during this event is apparent in Figure 8a but it is not explained by differences in the surface winds, since the maximum speeds in these (narrower) secondary jets are as large as those in the TWJ.

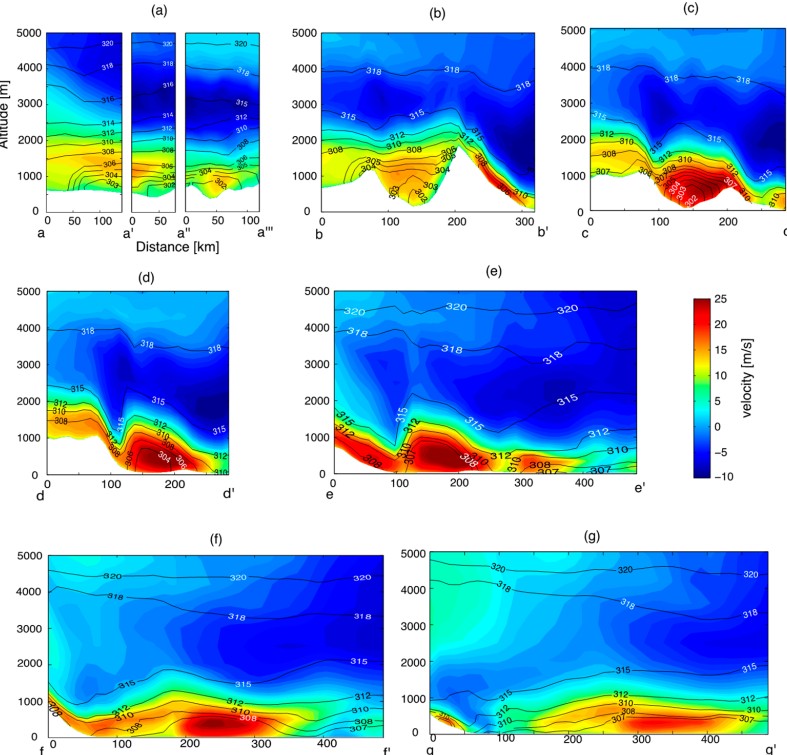

**Figure 7.** Cross sections of normal velocity and potential temperature, beginning upstream and proceeding down through the Tokar Gap and out over the Red Sea. The viewer faces downstream (roughly NNE). The data are plotted at 05 UTC (8 a.m. local time) on 12 July 2008. Section locations are shown in Figure 1.

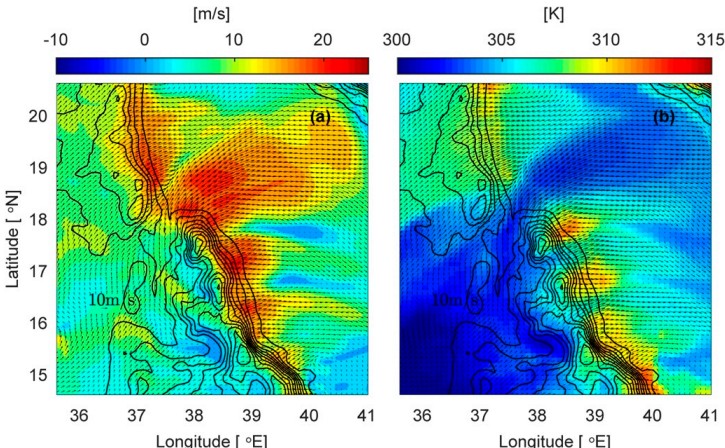

**Figure 8.** Plan views of 10 m wind speed (**a**) and ground-level potential temperature (**b**) at 05 UTC on 12 July 2008. Velocity arrows at 10 m are shown in both frames.

### 3.4. Mountain Parameter

There have been a number of modeling studies of gap winds in idealized settings, including [8] and [28–32]. A basic model [31] consists of a long ridge of height *h* and an upstream wind with uniform velocity *U* and buoyancy frequency *N*, approaching from the direction perpendicular to the ridge. A notch is cut through the ridge and thus an approaching air parcel can go around or over the ridge, or through the gap. The lowest elevation in the gap is the same as that of the surrounding flat terrain, so there is no sill.

One of the key parameters for flow over ridges with or without gaps is $Nh/U$. For values less than about 0.5, the flow tends to be inertial and the lower level air parcels ride directly up and over the topography in the manner of a hydraulically supercritical flow. For this regime, there is little tendency in the [31] model for the flow to divert around the ridge or through the gap, and consequently, a strong jet does not form in the gap. When $Nh/U$ exceeds a critical value, generally in the range of 0.7–1.1, for a ridge without a gap, breaking internal gravity waves begin to appear in the lee of the ridge crest [33]. The turbulence so generated can lead to the formation of a mixed layer or wedge of homogenized and relatively stagnant air. The wedge acts as a reflecting upper boundary for the flow below, which may then behave as a finite-depth layer that can undergo hydraulic transitions. This behavior was observed [31] in experiments with $Nh/U$ values of 1.4 and 2.8. There, breaking internal waves are observed in the lee of the ridge crest, away from the gap, and underneath these waves is a downslope flow that terminates in a feature resembling a hydraulic jump. Downstream of the jump the winds diminish considerably and a wake forms. A strong jet forms within the gap itself and this jet continues as a concentrated, narrow flow well downstream of the ridge. Slow, wake-like circulations consisting of counter-rotating gyres form on either side of the jet. When the $Nh/U$ is increased to the value of 5.0, there is an even greater tendency for the flow to be blocked by the ridge and much of it goes around. There is decreased flow into the gap and over the ridge top, and the strength and extent of the gap jet are diminished.

Estimation of $Nh/U$ for the Tokar Gap/Red Sea Hills region is difficult due to the nonuniformity of the upstream winds and the complexity of the topography. As shown in Figure 1, much of the terrain upstream of the gap (lower left portion of Figure 1) consists of a broad plain lying at about 400 m elevation. The elevation of the ridges and peaks that border the Tokar Gap ranges from 1200 to 2000 m, so a reasonable range of *h* is 800–1600, and we will use the average value of 1200 m. Wind profiles (Figure 9) taken at three locations within the upstream region at 00 UTC show winds directed towards the northeast (towards the gap) up to about 750 hPa (or about 2500 m above terrain) above terrain, above which they reverse direction. The average wind speed over this range is 6–8 m s$^{-1}$. The corresponding profiles of buoyancy frequency $N$ [$=g\,\theta^{-1}\,d\,\theta/dz$] have value $(0.012 \pm 0.002)$ s$^{-1}$ over much of this range but decrease to zero near the ground (Figure 9). We averaged these values from the ground level to 750 hPa in order to estimate the scales *U* and *N*. The resulting estimate of $Nh/U$, computed over the duration of the strong wind event (roughly 22 UTC on 11 July to 12 UTC on 12 July) varies between 1.0 and 4.0. For the model considered by [31], such values would suggest the existence of a strong gap jet with a long downstream extension as well as spilling flows across higher elevation ridges. However, the ridge in the model discussed by [31] lies at constant elevation, whereas the Red Sea Hills ridges are irregular and punctuated by peaks and cols, so there is little guidance as to where exactly spilling and lee wave breaking would occur. Moreover, there is no evidence of the weakly recirculating or stagnant 'wake' eddies that are seen in [31], possibly because the multiple jets in our case are too closely spaced to allow such features between them. The presence of the prevailing northwesterly winds along the axis of the Red Sea may also discourage closed circulation patterns.

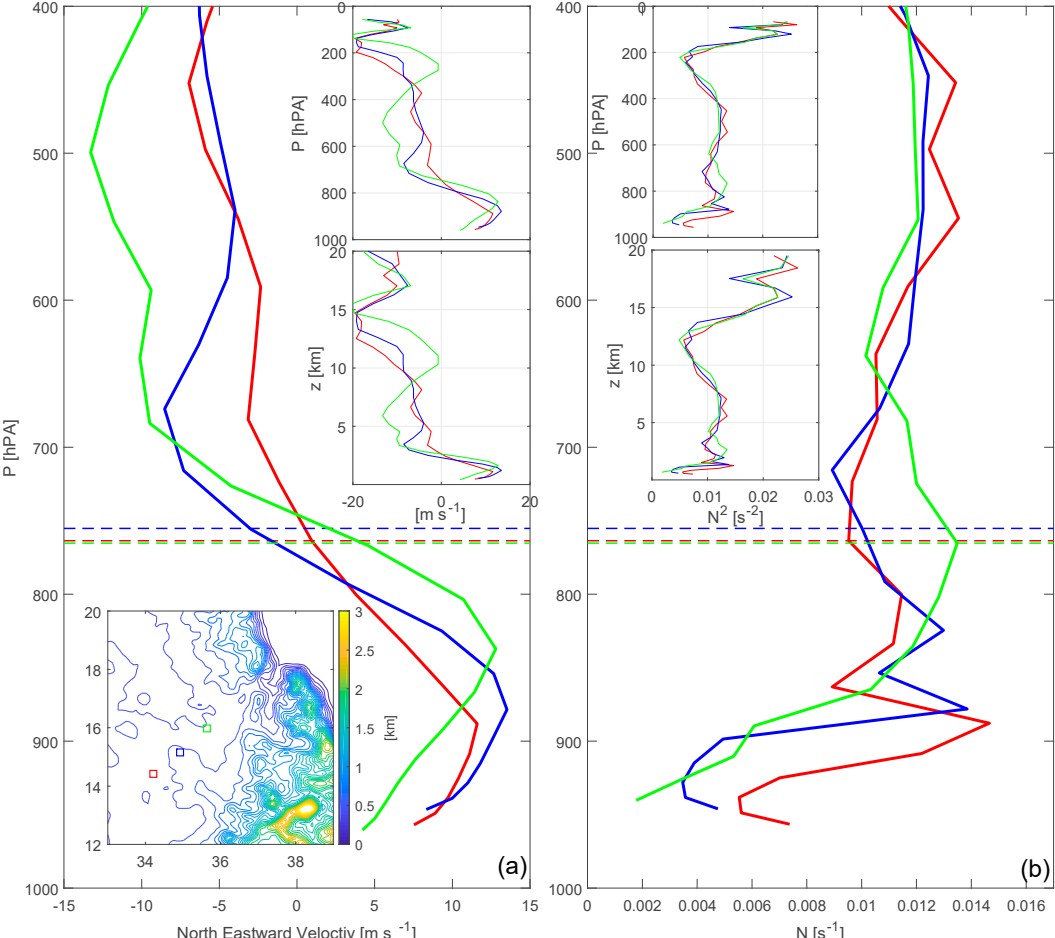

**Figure 9.** WRF profiles of the component of wind towards NE (**a**) and buoyancy frequency (**b**) taken at three stations upstream of the Tokar Gap at 7/12/2008 at 00 UTC. The station locations are indicated by the color boxes in the inset at the lower left of frame (**a**) showing topography contours. The $z = 2500$ m level, which roughly corresponds to the elevation of the highest peak in the Red Sea Hills, is indicated by a horizontal dashed line (one for each station.) The small inset profiles show the wind speed and $N$ over the full elevation range of the model, both in $z$ and in pressure.

## 3.5. Downstream Turning

Ref. [20] refer to the anticyclonic bending of dust plumes emanating from the Tokar Gap. As a result, the dust is transported southeast, along the axis of the Red Sea, sometimes covering the entire southern portion. This picture is consistent with the behavior of the 10 m winds (Figure 8a,b), which show the TWJ extending across the Red Sea towards the Saudi coast and then veering towards the southeast. The veering is even more pronounced for the jets in Gaps 2 and 3, although they do not extend as far offshore. The inertial radius $U/f$ based on a typical velocity $U = 15$ m s$^{-1}$ at 20° N is about 300 km, which is only moderately larger than the 200 km radius of curvature required for a southwesterly wind to veer anticyclonically 90° and flow towards the southeast before reaching the Saudi coast. Thus, the idea mentioned by [20] and others, that the turning is simply due to the Coriolis acceleration acting on the jets, has some support. However, bending may also be induced by the collision of the TWJ with the prevailing, geostrophic northwesterly winds that flow along the axis of the Red Sea. We also note that the Rossby number $U/fL$ for a $L = 150$ km wide jet with the above velocity scale is about 2, so the inertial character of the jets is significant.

### 3.6. Abrupt Transitions

In order to interpret some of the behavior cited above, it is helpful to make an analogy with the hydraulics of a single, homogeneous layer flowing beneath an inactive upper layer. The shallow-water analogy has been used in connection with other spilling mountain winds (e.g., [14]). Ref. [33] describes a number of conditions that would allow shallow-water/hydraulic interpretation to be valid. These include the presence of wave overturning aloft, which can produce a reflecting upper boundary, and the presence of a reflecting critical level. The features are certainly not ubiquitous in our model runs, but there is evidence of their presence at certain times and locations. For example, the wind reversals observed above the jet indicates that the along-axis flow changes sign at a level slightly above that of the spilling layer/jet (Figures 4–7). This level would also correspond to a critical level for stationary disturbances. In addition, the TWJ and jets in Gaps 2 and 3, though stratified, exhibit strong vertical coherence, as would a single, homogeneous layer.

As a bounding upper surface or interface of this hypothetical layer, we pick an isentropic surface (here 312 K) that roughly marks the top of the range of high, downslope velocities. The inspection of Figure 5 suggests that the 312 K surface is a reasonable choice, particularly over the high-velocity portions of the downstream flow. The resulting lower layer thickness $d$ (Figure 10a) suggests a pattern of spilling and thinning as air travels through the various gaps and out over the Red Sea. A significant difference is that the downslope flows in Gaps 2 and 3, and along the neighboring terrain, which contain air spilling from higher elevations, become quite shallow (dark blue) over the sloping terrain, but experience a rebound in layer thickness (light blue) as they move out over water. In contrast, the Tokar Gap outflow thins continuously as it spills out over the Red Sea, though it never becomes as shallow as the jets in Gaps 2 and 3. Note that the spilling air is not confined to the gaps but occurs broadly over the downslopes, as suggested in Figure 8a,b.

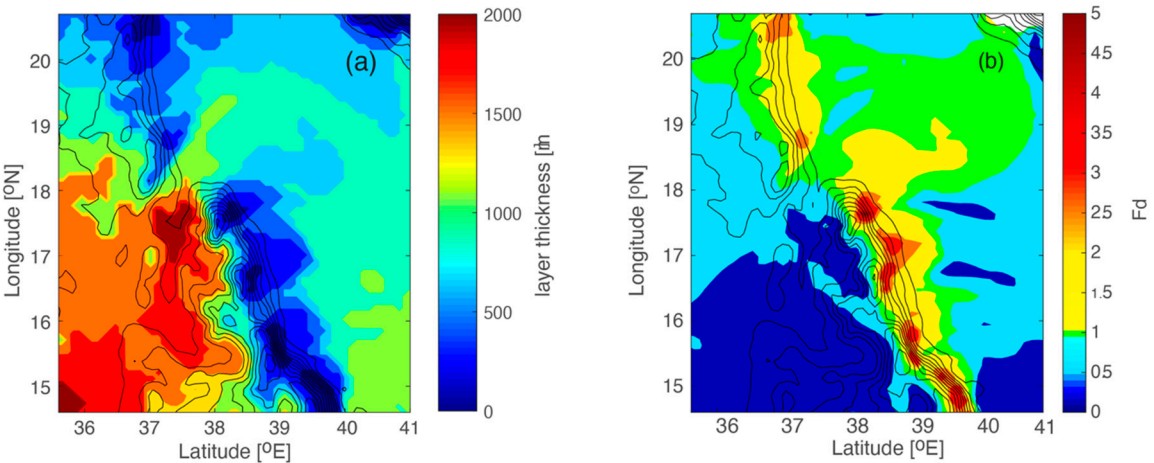

**Figure 10.** (**a**) Thickness of the layer formed between the ground level and the 312 K surface; (**b**) the local Froude number $F_d$ based on the 1.5-layer model with a interface at 312 K. Both plots for 05 UTC on 12 July 2008.

The local hydraulic state of the active layer of a 1.5-layer system is measured by the local Froude number:

$$F_d = \frac{U}{(g'd)^{1/2}} \tag{1}$$

where $g'$ is the reduced gravity $g(\overline{\theta}_2 - \overline{\theta}_1)/\overline{\theta}_1$ based on the average potential temperatures $\overline{\theta}_1$ and $\overline{\theta}_2$ below the interface and in the region of homogeneous potential temperature above the interface. In addition, $U = (\overline{u}^2 + \overline{v}^2)^{1/2}$ is the magnitude of the average horizontal velocity $(\overline{u}, \overline{v})$ over the layer. A plot (Figure 10b) of $F_d$ for the entire region, taken near peak wind conditions, suggests that the TWJ

does not undergo any significant hydraulic transition as the jet leaves the Gap. The Froude number of the exiting flow is close to unity and remains roughly so as the jet crosses the 200 km breadth of the Red Sea (green area).

Significant hydraulic transitions are indicated near the base of the Red Sea Hills, where $F_d$ reaches values that approach 5.0. An example is the jet that forms in Gap 2 immediately to the south of the TWJ and that spills from a pass of elevation 1360 m lying between the two peaks of above 1500 m. As shown by a longitudinal section of this flow (Figure 6), the overlying 315K contour descends down the lee slope but suddenly rises at the base of the slope (near 150 km). For a 1.5-layer flow, the relative change in layer thickness across a jump is given by

$$\frac{d_1}{d_0} = \frac{\sqrt{1 + 8F_o^2} - 1}{2} \qquad (2)$$

where $F_o$ is the value of $F_d$ just upstream of the jump and $d_0$ and $d_1$ are the values of layer thickness just upstream and downstream of the jump, respectively [34]. This formula expresses the general tendency for relative change $d_1/d_0$ to become larger as the upstream Froude number increases, but it ignores features such as bottom friction, entrainment and continuous stratification. With $F_o$ and corresponding $d_0$ values that range between (2.5, 250 m) and (5.0, 100 m) for winds emanating from Gaps 2 and 3, and long neighboring slopes, the predicted downstream thickness $d_1$ is 650–770 m. The downstream thickness values apparent in Figure 10a (light blue to light green) lie in the range of 700–1000 m. It is therefore reasonable to conclude that observed transitions have some similarity to classical hydraulic jumps, though this assertion comes with the caveat that the WRF model resolution may be insufficient to capture rotors and other detailed structures associated with observed downslope wind transitions (e.g., [14]).

### 3.7. Downstream Extent

The presence of hydraulic transitions in the jets emanating from Gaps 2 and 3, and the absence of such in the Tokar Wind Jet, suggests that the latter might have much a longer downstream extent than the former, as observed. For one thing, the TWJ would not suffer the energy dissipation that occurs in a hydraulic jump, and that is proportional to the cube of the upstream/downstream difference in layer thickness. Another quantity that favors the TWJ in terms of downstream reach is the offshore momentum flux. For a 1.5-layer, unidirectional flow, the momentum flux (or total 'flow force') per unit breadth of flow is given by

$$M = g'd^2 + \bar{u}^2 d \qquad (3)$$

where $\bar{u}$ is the offshore component of the vertically averaged velocity over the layer. With estimated values of the layer thickness $d$ and velocity along the centerlines of the TWJ and Gap 2 jet, we find $M$ in the range of $(6–10) \times 10^5$ m$^3$ s$^{-2}$ for the former, and $(2–3) \times 10^5$ m$^3$ s$^{-2}$ for the later. So despite the fact that the offshore velocities in the jets emanating from Gaps 2 and 3 can be larger than those in the TWJ, the latter carries two to three times larger total momentum flux than these secondary jets.

## 4. Lagrangian Structure

To obtain a more comprehensive view of the Lagrangian structure of the winds in and around the Tokar Gap, we initiated groups of trajectories in the high-speed outflow regions and integrate backward and forward in time. For example, Figure 11 shows backward-time (red) and forward-time (black) trajectories initiated at 100 m elevation and within a small horizontal region lying close to where the TWJ crosses the coast (the region is formally defined as that where the wind speed exceeds a threshold value, here 15 m s$^{-1}$). The trajectories are integrated from the initiation time (also labeled on each frame) backward to 12 UTC on the previous day (11 July 2008), and forward to 24 UTC of the current day (12 July 2008), thus revealing information about upstream sources and downstream extent

of the wind event. Frames a–d, which show the results of releases at 00, 03, 06 and 09 UTC, suggesting that while some of the flow is fed upstream by low-level winds over the plateau to the southwest of the Tokar Gap, there is also a significant contribution from an isolated feature containing descending air parcels. We will refer to this feature as the *upstream cell* and discuss it in more detail below. Examples of these trajectories that descend from the cell are colored green and their upstream origins are indicated by arrows. As for the downstream (black) segments, it is apparent that the majority of air parcels cross the Red Sea and come close to the Saudi coast, some penetrating inshore and up and over the Saudi coastal mountain range (Frame a). This penetration is notable given the excess moisture content of such parcels, as described by [6].

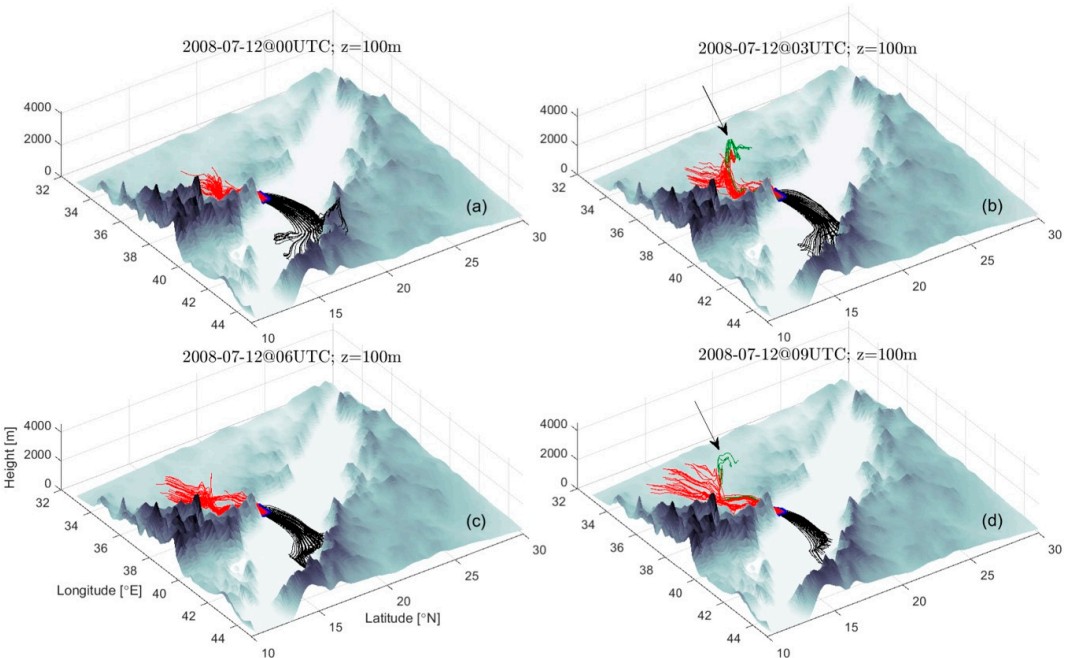

**Figure 11.** Forward time (black) and backward time (red) trajectories, initiated at the time indicated on each frame and within a horizontal patch (blue area) lying at $z = 100$ m and where the wind speed exceeds 15 m/s. Trajectories are integrated backward to 12 UTC on 12 July 2008 and forward in time to 24 UTC on 12 July 2008. The arrows point to trajectories that originate from the upstream cell containing strong downdrafts. These trajectories are colored green.

A better view of the vertical Lagrangian structure of the TWJ appears in Figure 12, where the viewer now faces southeast. Each frame shows four groups of color-coded trajectories initiated at levels of 100, 500, 1000 and 1500 m, and the release times are as in Figure 11. It can be seen that the funneling winds are fed by a horizontally broad collection of trajectories that cover the upstream plane. Trajectories released at higher levels, colored yellow and magenta, tend to originate from the north portion of the upstream plateau (the portion closer to the viewer). Interestingly, trajectories released at lower elevations (dark red and bright red) predominantly originate from the southern portion of the plateau and make up the bulk of the air parcels that descend from the isolated cell mentioned in the previous paragraph. A small number of the trajectories originate at higher elevation above the Red Sea and move westwards before descending and reversing their direction as they flow down into the gap. The forward (black, blue, green and cyan) trajectories all cross the Red Sea, and most of the ones released at lower elevation (predominantly black, blue and some green) cross the Saudi coast and penetrate inland.

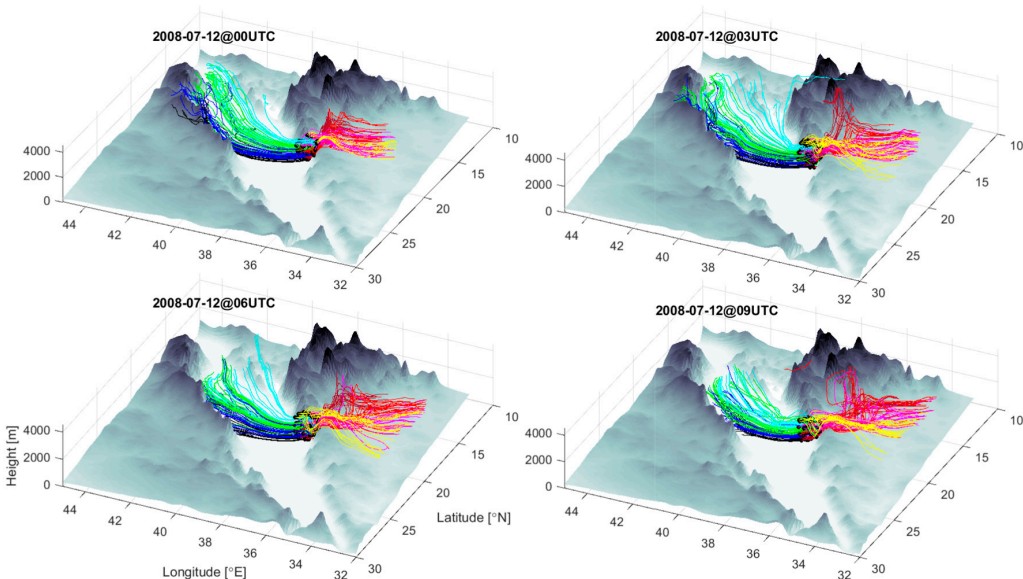

**Figure 12.** Similar to Figure 11, but now the viewer faces SE and the trajectories are initiated at the four levels *z* = 100, 500, 1000 and 1500 m and where the wind speed exceeds 15 m/s. The color coding is 100 m: (dark red to black for backward-to-forward in time); 500 m (bright red to blue); 1000 m (magenta to green); and 1500 m (yellow to cyan). A few trajectories appear to be discontinuous because their paths took them above the upper boundary of the plot.

The wind jets that form in Gap 2 and Gap 3 exhibit some Lagrangian characteristics that are distinct from those of the TWJ. The winds in Gap 2 spill over a relatively high and narrow sill at 1360 m elevation (Figure 6). Trajectories initiated within the outflow region at 100 m (not shown) show no connection to the upstream cell, while those initiated at (or above) 500 m (Figure 13a–d) exhibit a strong connection. In addition, none of the trajectories cross the Red Sea but instead turn southwards and flow parallel to the coast, some eventually crossing back into Africa (Frames a–c). A magnified and rotated view (Figure 13e) of frame c more clearly shows that after air parcels leave the coast, they experience a rapid ascent as they pass through the suspected hydraulic jump, then turn rapidly towards the southeast. Trajectories in Gap 3 (not shown), which is broader than Gap 2 and has a higher elevation (1430 m) sill, exhibit similar features. The upstream region in Figure 13e contains a number of trajectories that descend in a nearly vertical alignment, but there are also neighboring trajectories that experience temporary ascent before descending into the overflowing gap wind.

The upstream region of descending air parcels bears resemblance to a feature identified by [6] in connection with a separate strong wind event in the Tokar Gap (see the 'downburst' in their Figure 9c). The event in question began during the late evening of 8/12 and extended into 8/13. Videos of the two events (see WRFVel_0711 and WRFVel_0712 in Supplemental Information) reveal that in both cases the feature is a coherent cell of negative vertical velocity surrounded by distinct patches of rising air, and that these disturbances form near the upstream entrance to the Tokar Gap at about the same time as the gap winds begin to blow. The videos also show that the features propagate westward, covering approximately 300 km in 10 h in each case, and eventually passing out of the high-resolution subdomain. As noted by [6], the northern edge of the monsoon air mass is also aligned in the east–west direction and the features approximately follows this edge as they move to the west.

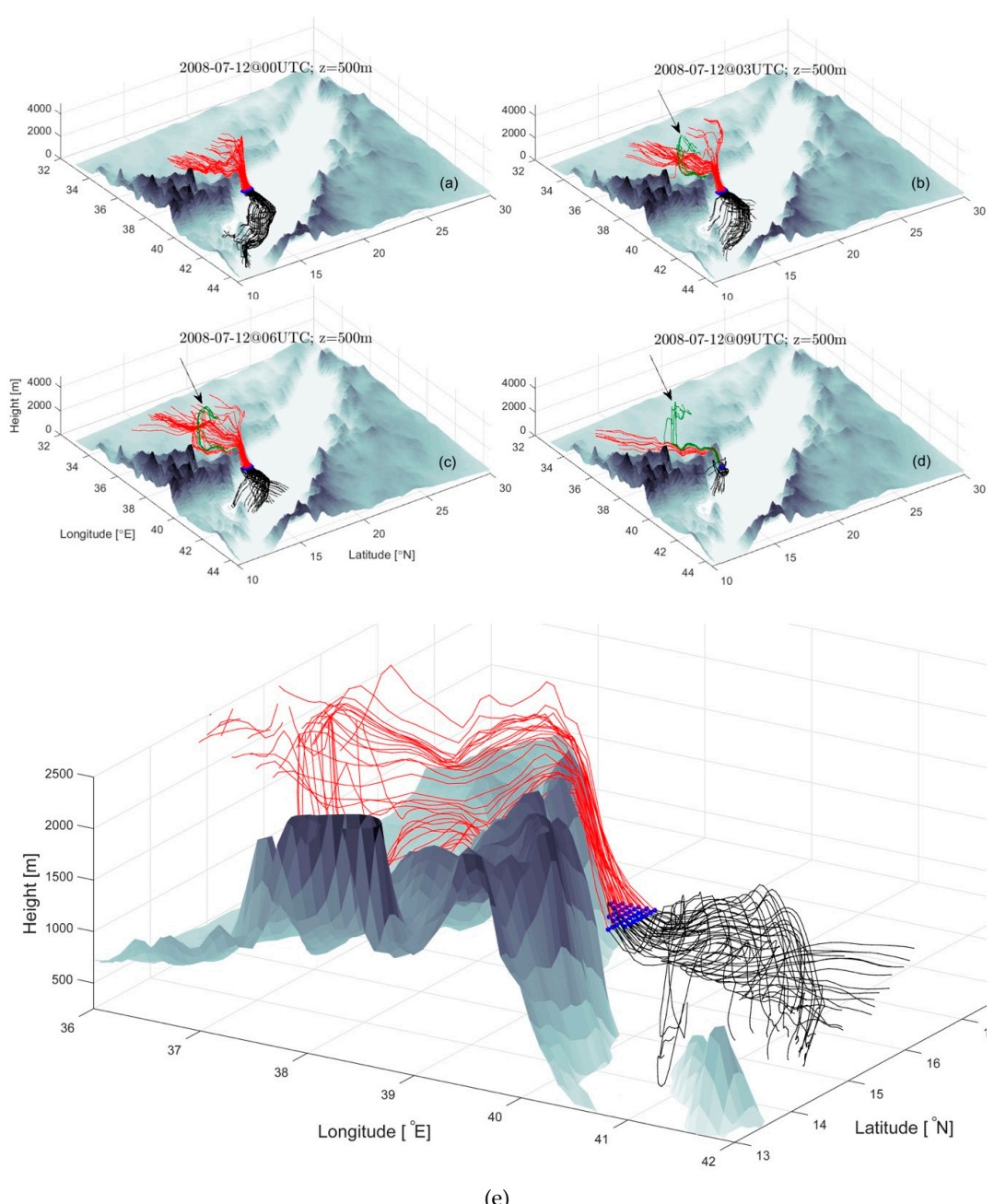

**Figure 13.** Similar to Figure 11, but the trajectories are released at *z* = 500 m in the outflows of the Gap 2 (the gap immediately to the south of the Tokar Gap) over an area where the wind speed exceeds 15 m/s. As in Figure 11, arrows point to (green) trajectories that originate from the upstream cell. Frame e is an enlarged and a slightly rotated version of frame **c**. The upstream portions of some trajectories appear discontinuous because they spent time above the upper boundary of the plot domain.

The connection between the upstream cell of descending air and the jet outflows is further illustrated in Figure 14, where trajectories are initiated at 1000 m (frames a–c) or 1500 m (frames d–f) and within the core of the cell (defined as the horizontal area in which the downward velocity exceeds 0.25 m s$^{-1}$). Backward-time (red) segments show that some of the air parcels descend from as high as 3000–4000 m, while others can be traced upstream along more horizontal paths. The upstream drift of the cell as a whole is also apparent over the elapsed time of 6 h. The forward-time (black) segments all pass downstream through the Tokar Gap proper when initiated at 1000 m, while some trajectories initiated at 1500 m enter Gaps 2 and 3.

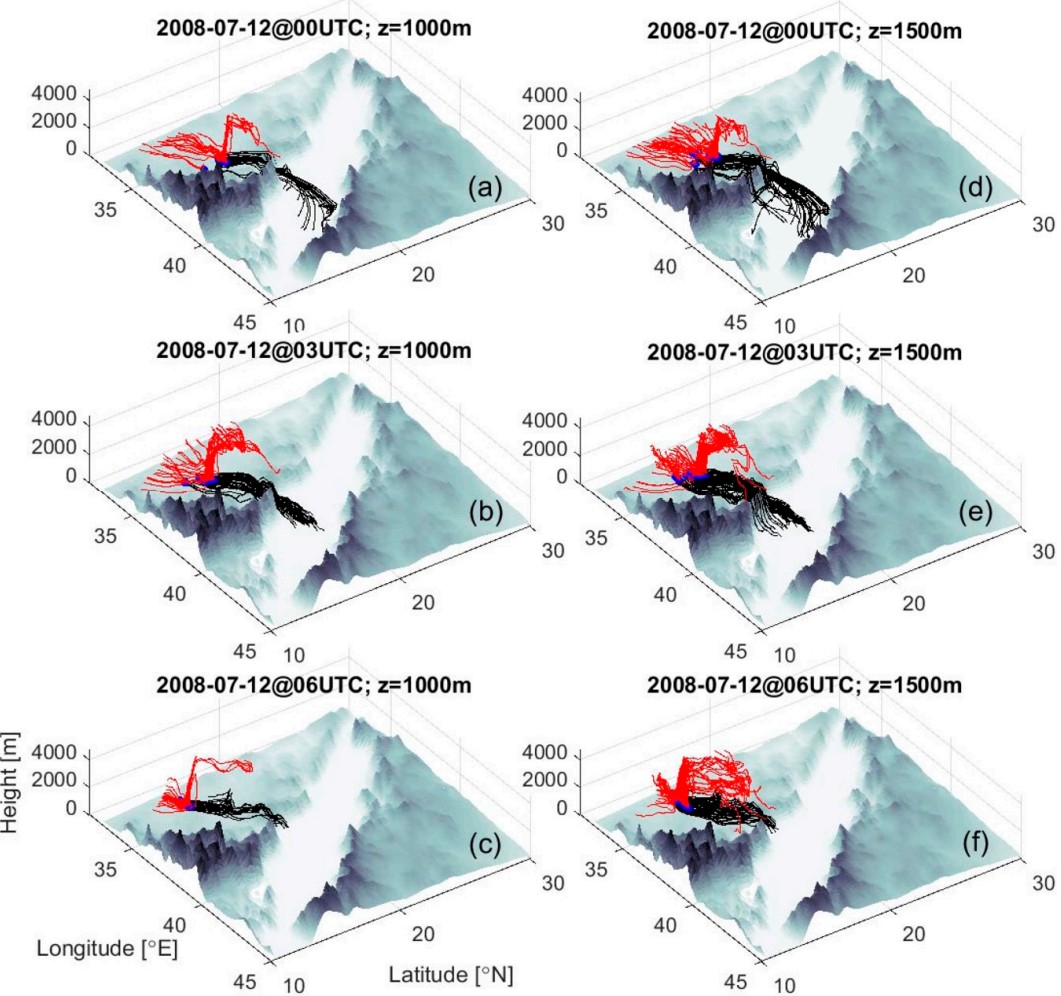

**Figure 14.** Trajectories initiated within the upstream cell at $z = 1000$ m (frames (**a**–**c**)) and 1500 m (frames (**d**–**f**)) at the times indicated and over areas where the downward vertical velocity, $w$, exceeds the value 0.25 m/s.

The Eulerian structure of the cell is partially revealed by horizontal maps of the vertical velocity at $z = 5500$ m at consecutive times, as shown by the top row of panels in Figure 15. Although the downward motion (blue) is indeed present, there is an accompanying patch of upward motion (yellow), in addition to other yellow patches in the region, all drifting westward. Vertical sections though the feature (middle row) show that the accompanying patches of rising and descending motion exhibit a strong degree of vertical coherence. Vertical sections of the vertical component of relative vorticity $\zeta = \frac{\partial v}{\partial x} - \frac{\partial u}{\partial y}$ indicate a predominance of anticyclonic vorticity at lower levels, with cyclonic vorticity at higher levels. [6] described the feature as a cyclonic cell, but Figure 15 shows a more complex structure.

Energy transformations experienced by air parcels along the paths of the jets are revealed by an examination of the terms that compose the Bernoulli function $B$ for compressible flow [35]:

$$B = e + \frac{p}{\rho} + \frac{|\mathbf{u}|^2}{2} + gz = c_v T + \frac{p}{\rho} + \frac{|\mathbf{u}|^2}{2} + gz \tag{4}$$

where $e$ is the specific internal energy. For a dry gas, $e$ is equal to the product of the specific heat $c_v$ (assumed to have constant value 171 J/(kg K)) and the in situ temperature $T$. In a steady, adiabatic and isentropic flow, $B$ is conserved along fluid trajectories. While these conditions do not generally hold in our applications, the winds during the strong phase of the 12 July 2008 event are approximately

steady and as we will show, the value of *B* undergoes only slight variation for an air parcel descending through the Tokar Gap or neighboring gaps during the duration of the strong event. In addition, the variation of *B* during this phase can be shown to be primarily due to diabatic heating.

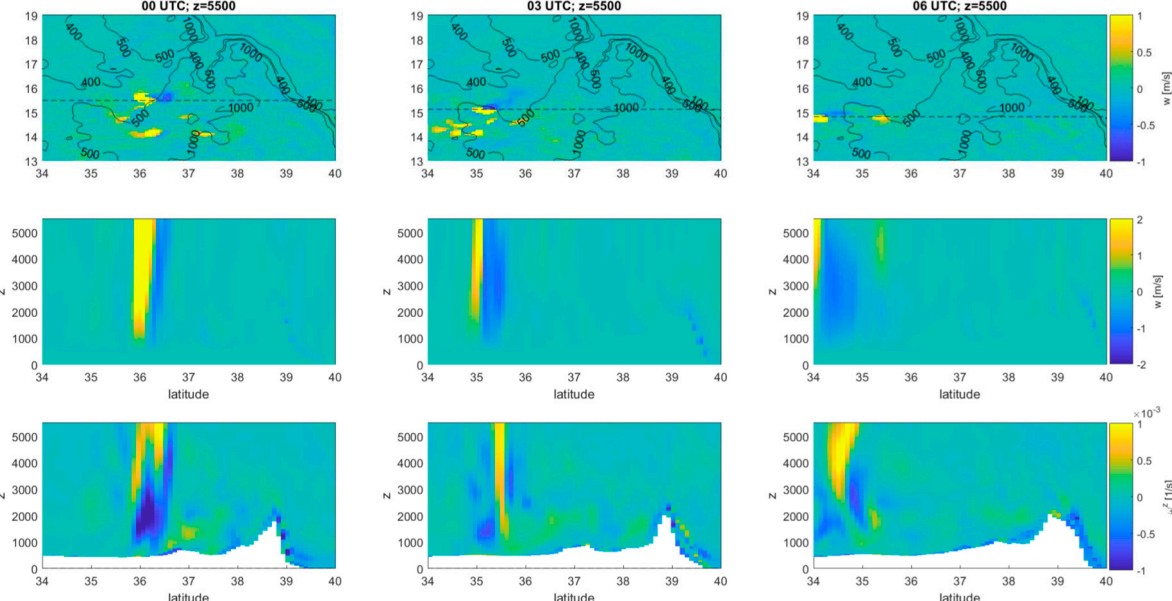

**Figure 15.** Plan views of vertical velocity, $w$, at $z = 5500$ m (top row); vertical sections of $w$ (middle row), with the section positions indicated by the dashed lines in the upper panels; vertical sections of the vertical component of relative vorticity $\zeta$ (bottom row) over the same sections. The three columns of panels are taken at the times indicated.

We chose six trajectories (Figure 16f) along which to track changes in B and its constituents. Three of the trajectories pass through the Tokar Gap and three pass through Gap 2. The former are colored pink upstream of, and green downstream of, the release locations (blue dots) near the coast. The trajectories passing through Gap 2 are colored red/black. The time histories of *B* and its constituents for each trajectory are tracked in the left-hand panels of Figure 16.

For all trajectories, *B* experiences only small and gradual changes (Figure 16a) during the 8 h period before release and the 4 h period after release, which covers most of the duration of the event. However, each experiences an abrupt increase in *B* right around the release time at $t = 18$ (06 UTC, or 09 LST). At this time, the five air parcels descended to the coastal plain and are about to move out over the water. The increase in *B* can be attributed to diabatic heating, as evidenced by an increase in potential temperature (Figure 16g). In fact, the similarity between the curves in panels (a) and (b) suggest that it is diabatic heating/cooling, and not time-dependence, that is primarily responsible for the changes in *B*.

The plot of potential energy $gz$ (Figure 16d) shows that the Tokar Gap trajectories (in pink) experience a gradual and nearly monotonic descent as they pass through the gap and out over water during the first 6 h of the strong wind event, whereas the Gap 2 trajectories (in red) ascend and then descend rapidly as they move up and over a ridge at much higher elevation. The Gap 2 trajectories also experience an abrupt rebound (at about $t = 8$) shortly after they have moved out over water (see black extensions of red curves). This rebound coincides with the suspected hydraulic jump. Panel *e* shows that the kinetic energy of the air parcels in the Tokar Gap outflow remains high for 3 or 4 h after the parcels have left the coast, whereas the Gap 2 trajectories experience a sudden decrease in kinetic energy at the positions of the jumps. Although the changes in kinetic and potential energy meet expectations for a hydraulic jump, the value of *B* itself does not experience any notable change. As shown in Frame g, the jump occurs during the period when the air parcels are experiencing the strongest periods of diabatic warming, and this may compensate for the dissipation of kinetic energy within the jump

(a process that is poorly resolved by the model). We also note that [36] have documented examples of stratified jumps that exhibit little or no energy dissipation. A model with higher horizontal and vertical resolution may ultimately be needed to clarify this picture.

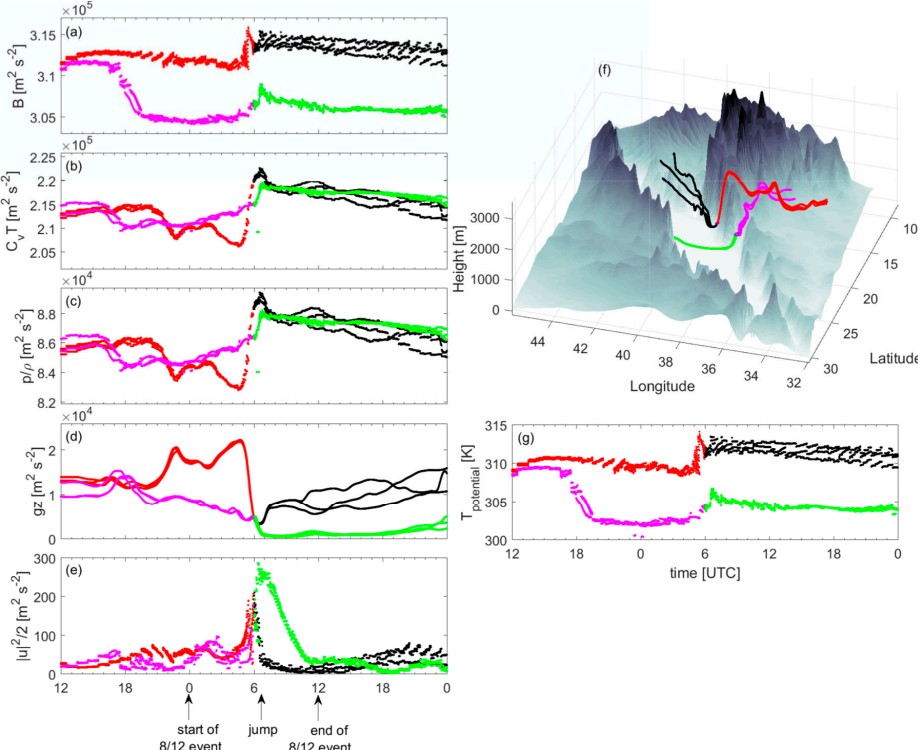

**Figure 16.** The Bernoulli function (**a**) and its constituents (**b**–**e**) as functions of time (UTC) following the six trajectories plotted (**f**). The three pink/green trajectories pass through the Tokar Gap while the three red/black trajectories pass through Gap 2. The trajectories are initiated at the (blue) transition point and integrated back and forward in time. Frame (**g**) shows potential temperature along the four trajectories. The unevenness of some of the curves is a result of the interpolation method.

Overall, the most significant constituents of $B$ are the pressure term $p/\rho$, the potential energy $gz$ and the internal energy $c_v T$ (Figure 16, panels b–d). As the air parcels descend over the topography, $c_v T$ and $p/\rho$ both increase at the expense of $gz$. Interestingly, although the kinetic energy increases, it does not contribute significantly to the balance. This is in sharp contrast to deep overflows in the ocean (Pratt and Whitehead 2008), which are nearly incompressible and experience only minor changes in internal energy, and where the kinetic energy increase plays a more substantial role in the overall budget.

Although the flow in Gaps 2 and 3 likely experience hydraulic jumps, the overall horizontal spread of air parcels is weaker than for the TWJ. To quantify particle spreading, we computed the single-particle dispersion tensor:

$$D_{ij} = \frac{1}{N} \sum_{n=1}^{N} \left( dx_i^n - \overline{dx_i} \right)\left( dx_j^n - \overline{dx_j} \right)$$

where $i, j = 1, 2, 3$ correspond to the Cartesian coordinates $(x, y, z)$, overbars denote an ensemble mean, and $dx_i^n = x_i^n(t) - x_i^n(0)$ is the displacement of an n-th parcel from its initial position. The dispersion matrix can then be put in a diagonal form, with the three eigenvalues, $D_\tau$, $D_n$, $D_z$, on the diagonal (since the horizontal velocity is generally much larger than the vertical velocity, the largest first two eigenvalues approximate the horizontal dispersion; the third approximates the vertical dispersion,

something which we verified by inspection). A comparison (Figure 17) based on these three coefficients between trajectories released at the exits of the Tokar Gap (left panels) and Gap 2 (right panels) yields some striking differences. At each location, a group of 25 air parcels is released every hour from 00 UTC until 09 UTC at $z = 500$ m elevation at the exit and where velocity exceeds 15 m/s. The horizontal components of dispersion, $D_\tau$ and $D_n$ for the trajectories released in Gap 2 grow at a slower rate than those for an equivalent group of parcels released at the exit of the Tokar Gap. The difference is not surprising given the large spread of trajectories originating in the Tokar Gap (top panels of Figure 17). The vertical dispersion $D_z$, on the other hand, is slightly larger for the Gap 2 parcels during the initial 6 h since particle release and until TWJ parcels reach the opposite coast and start rising up over the mountain ranges, at which time the vertical dispersion for the TWJ parcels becomes larger. The same is true for parcels released at 100 m (not shown). In both cases the dispersion is dominated by the horizontal spreading of the trajectories, which is more pronounced in the TWJ.

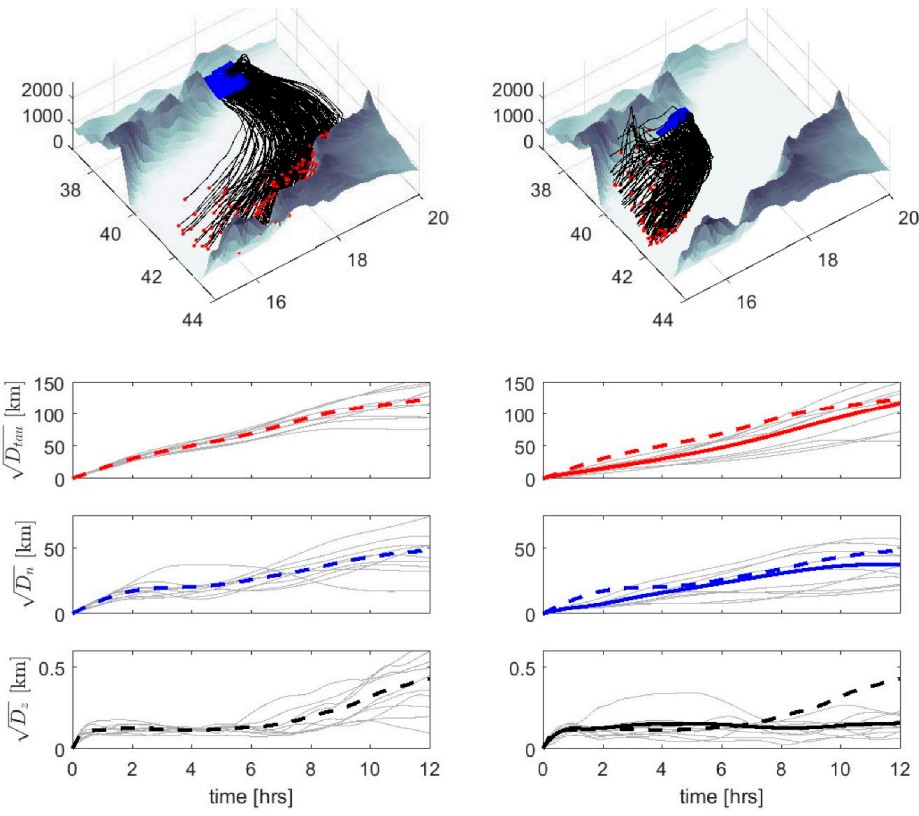

**Figure 17.** Single-particle dispersion ($D$) in the horizontal ($D_\tau$, $D_n$) and vertical ($D_z$) direction calculated for groups of trajectories initiated at the exits of the Tokar Gap (left) and Gap 2 (right). Twenty-five trajectories were released every hour from 00 UTC to 09 UTC on 07/12/2008. Dispersion curves for the individual releases are shown in grey, and means are shown by thick colored curves, dashed for the Tokar Gap and solid for Gap 2. Dashed curves are superimposed in the right panels for comparison purposes. In each case, the trajectories were initiated at $z = 500$ m, and where the horizontal wind speed exceeds 10 m s$^{-1}$.

## 5. Discussion

The strong wind event discussed in this paper exhibits structural similarities with several other strong wind events during July and August (not shown) in the WRF model. Peak winds in the Tokar Gap during those events were 20 m s$^{-1}$ or less, and although the wind jets extended far out over the Red Sea, they exhibited stronger veering to the southeast and did not always reach the Saudi coast. The increased veering is consistent with a decreased inertial radius $U/f$ and also with a greater influence of the ambient northwesterly winds blowing along the axis of the Red Sea. During the other events,

the layer thicknesses based on the 312 K surface are generally smaller than during our 12 July 2008 extreme event, and the local Froude number distributions show marginally supercritical flow within the Tokar Gap and supercritical flow within Gaps 2 and 3. As in the extreme case on 12 July 2008, the jets in Gaps 2 and 3 experience an increase in layer thickness and drop in speed, possibly due to hydraulic jumps, as they blow out over the Red Sea. Finally, we found no evidence of the upstream cell with a core of descending motion that was described above and also occurred during the 7/13 wind event [6].

The Tokar Delta is often cited as a major source of summer dust plumes that can cover significant portions of the Red Sea and Arabian peninsula [20]. In the absence of a full aerosol model, there are several 'rules of thumb' that meteorologists commonly use to determine whether conditions are favorable for the lofting of dust up into the atmosphere (see www.meted.ucar.edu/mesoprim/dust/print. htm). We now examine those criteria at a site where the core of the wind jet crosses the Tokar Delta region (indicated by a triangle close to the Red Sea shoreline in Figure 1) over the 24 h span of extreme wind event (Figure 18). The first condition was that the ground-level wind speed exceeds 7–8 m s$^{-1}$, which is usually satisfied when the winds at 1000 ft. exceed 15 m s$^{-1}$. It can be seen from Figure 18 that both conditions are satisfied during all but the evening period (13–18 UTC) when the wind relaxes. [37] describes the lifting of dust at the leading edge of the monsoon flow, a phenomenon that could be relevant during the onset of the TWJ, when the leading edge of the cool and moist monsoon air mass passes down through the Tokar Gap. Although the wind jets in Gaps 2 and 3 achieve near-surface velocities strong enough to lift fine silt, these gaps do not have comparable delta regions.

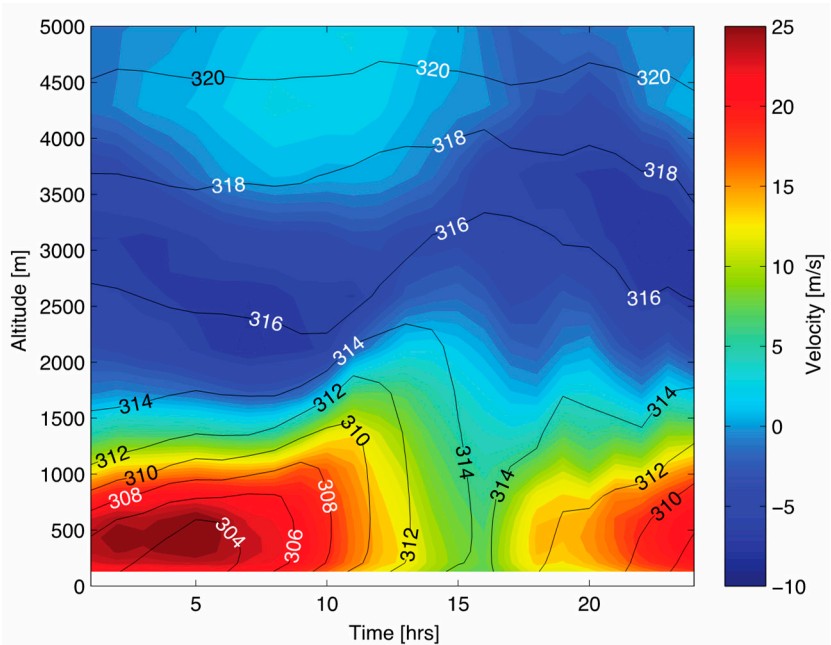

**Figure 18.** Hovmöller diagram for potential temperature and horizontal, along-thalweg wind velocity as a function of elevation over 12 July 2008. The data come from a location over the Tokar Delta, indicated by a triangle in Figure 1.

Even though the winds may be strong enough to lift dust, unstable stratification is required for the dust to be lofted up to 1000 m. Figure 18 indicates that the TWJ is capped by stable stratification in the 307–312 K range, roughly from 500 to 1000 m, during its strongest phases (00–10 and 21–24 UTC). However, the region below this transition layer is relatively homogeneous in potential temperature, and statically unstable near ground level, so it is possible that dust could be lofted up to fill a column of 500 m above ground. During the weak phase of the TWJ, potential temperature is well mixed up to about 1500 m, but the winds are too weak to lift the dust off the ground. Previously suspended dust

within 500 m of ground could be lofted higher during an afternoon weak phase by thermal convection. A similar example over the Sahara is discussed by [38].

## 6. Conclusions

The main thrust of this work was to describe the anatomy and dynamics of a simulated strong wind event in the Tokar Gap and neighboring gaps, to offer an explanation for the relatively limited downstream reach of the wind jets in the neighboring gaps, and to describe a Lagrangian overview of the circulation associated with the jets. The analysis suggests that the Tokar Wind Jet (TWJ), which spills down through a ravine-like topography, never achieves supercritical speeds and therefore does not undergo a dissipative hydraulic jump as it departs the gap. By contrast, the jets in the neighboring gaps form when air spills down from relatively high passes, resulting in a layer of air that achieves comparable velocities to those of the TWJ but is shallower and therefore more supercritical. These jets do, in fact, experience features resembling hydraulic jumps after they depart the coastline and become much thicker and weaker downstream of the jump. The model resolution is inadequate to resolve the detailed features of the presumed jumps, but the layer thickness increases, and the horizontal velocity decreases, as air parcels cross it. Downstream of the jump, air parcels turn quickly to the southeast and flow in that general direction, never crossing the Red Sea. The downstream extension of the TWJ easily crosses the Red Sea and reaches the Saudi coast, with some air parcels penetrating across the coastal mountain range and further inland. The horizontal spread of particles, as quantified by the single-particle dispersion tensor, is larger for the Tokar Wind Jet than that for the neighboring gaps. The vertical spread is, on the other hand, slightly larger for the secondary gaps during the first six or so hours, plausibly due to the effects of the suspected hydraulic jumps.

The energetics of all the jets, as quantified by transitions in the various terms that constitute the Bernoulli function, suggest that the primary exchange is between $p/\rho$, potential energy, and internal energy, with kinetic energy changes playing a surprisingly secondary role. Diabatic heating along the coast increases the Bernoulli function and makes it difficult to isolate any dissipation of the energy associated with the jumps.

The above scenario has many elements in common with idealized models of ridges with single gaps when the mountain parameter $Nh/U$ is comparable. This includes a gap jet with a long downstream extension, and a supercritical flow that spills down from the high-elevation ridge crests and terminates in a hydraulic jump (note, however, that idealized models typically do not include secondary gaps). The lofting of dust into the atmosphere, as observed frequently over the Tokar Delta, is consistent with threshold wind values in our WRF model.

Our analysis revealed another interesting phenomenon, one that was documented by [6] as part of a separate strong wind event and that is not easily explained by earlier work. During the onset phase of the 12 July 2008 event, a coherent cell with strong descending air and nearby patches of ascending air is generated near the upstream entrance of the Tokar Gap. Some of the air parcel trajectories that enter the TGJ as well as Gap 2 emanate from this cell. As the jet evolves and eventually weakens, the cyclonic cell moves to the west and out of the high-resolution domain. This feature is not observed for any other events occurring in July and August of that simulation year. Whether strongest events are made so by the presence of the cell, or vice-versa, is a subject that invites further analysis.

**Supplementary Materials:** The following are available online at http://www.mdpi.com/2311-5521/5/4/193/s1, Video S1: WRFVel_0711, Video S1:WRFVel_0712.

**Author Contributions:** L.J.P. conceived the project, wrote the paper and helped with the construction of many of the figures. He supervised E.J.A., who performed the Eulerian analysis and constructed most of the figures showing Eulerian properties during a summer at the Woods Hole Oceanographic Institution as a visiting student. I.R. performed all of the Lagrangian analysis, constructed all the figures showing trajectories and dispersion, performed the Bernoulli analysis, constructed Figure 15, and made some improvements in other figures. H.J. performed the numerical calculations. All co-authors proofread different versions of the manuscript. All authors have read and agreed to the published version of the manuscript.

**Funding:** This study is part of a joint research project by King Abdullah University of Science and Technology (KAUST) and the Woods Hole Oceanographic Institution and was funded by KAUST. Support for Albright was provided by the National Science Foundation under Grant (OCE-0525729). Additional support for Pratt came from KAUST through award CRG6-2017-3408.

**Acknowledgments:** The authors are grateful to Stephen Maldonado, who assisted with the estimates of the mountain parameter, and to Shannon Davis for some helpful comments.

**Conflicts of Interest:** The authors declare no conflict of interest.

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
