# Peer review of "Eulerian and Lagrangian Comparison of Wind Jets in the Tokar Gap Region"

_fluids, doi:10.3390/fluids5040193_

Round 1

Reviewer 1 Report

Please see the attached file with my comments.

Reviewer 2 Report

Dear editor:

The authors have studied the Eulerian and Lagrangian Comparison of Primary and Secondary Wind Jets in the Tokar Gap Region is presented in this article. Tokar gap and the other two unnamed gaps in the region generate strong wake flows towards the Red sea and hence contribute greatly to dust storms and material transport; therefore, the problem is one of practical interest. The research methodology is sound and the conclusions are consistent with physical observations. The reviewer, therefore, accepts the manuscripts for publication with the following minor corrections:   

  1. In line 83 it is stated that:

“The fields shown are means for July 2008 and the wind jets that form are not as distinct as they would be in instantaneous examples (e.g. Fig. 8 below) but the mean TWJ can clearly be seen near 18oN and the mean expressions of several other gap flows can be seen to the south.”

Are these simulation results validated with any other source of data, be it expertly easements or satellite images? It will be helpful if some tangible results are presented, instead of referring to other studies (given the free access nature of this journal).

  1. In line 130 we read:

“Our results are based on 14-month run of the Weather Research and Forecasting (WRF) model, version 3.0.1.1, with a 10-km horizontal resolution Red Sea subdomain nested within a 30-km resolution domain covering most of the Middle East, for which the 1° NCEP Global Final Analysis was used as initial and boundary conditions”

Given the multiscale nature of the problem (i.e. the effect of spatially narrow gaps on the overall flow pattern in a very large field) and rapid transitions in time and space that are reported by the authors (i.e. hydraulic transition), the reviewer is not convinced this is a sufficiently fine grid. It will be helpful to provide convergence studies and build confidence around this matter.

  1. In line 440 we read:
    “The upstream region of descending air parcels bears resemblance to a feature identified by Davis et al. (2015) in connection with a separate strong wind event in the Tokar Gap (see the ‘downburst’ in their Fig. 9c).”

Please provide plots or graphs to support this claim.

  1. For Lagrangian transport studies, it seems that only air particles are studied. Given that In Figure 2, the dispersion of dust particles is depicted, it is not clear to the author as to why dust particles of several dimeters are not chosen to use this image as a validation reference (at least in terms of traveling distance and drift velocities). If it is a heavy task, please consider them to include these studies in your future publications.
